DATA RELEASE

# Chromosome-level genome assemblies of five *Sinocyclocheilus* species

Chao Bian[1,†], Ruihan Li[2,†], Yuqian Ouyang[1,†], Junxing Yang[3], Xidong Mu[4,*] and Qiong Shi[1,5,*]

1 Laboratory of Aquatic Genomics, College of Life Sciences and Oceanography, Shenzhen University, Shenzhen, Guangdong 518057, China
2 CAS Key Laboratory of Mountain Ecological Restoration and Bioresource Utilization & Ecological Restoration and Biodiversity Conservation Key Laboratory of Sichuan Province, Chengdu Institute of Biology, Chinese Academy of Sciences, Chengdu, Sichuan 610041, China
3 State Key Laboratory of Genetic Resources and Evolution, The Innovative Academy of Seed Design, Yunnan Key Laboratory of Plateau Fish Breeding, Kunming Institute of Zoology, Chinese Academy of Sciences, Kunming, China
4 Key Laboratory of Prevention and Control for Aquatic Invasive Alien Species, Ministry of Agriculture and Rural Affairs, Guangdong Modern Recreational Fisheries Engineering Technology Center, Pearl River Fisheries Research Institute, Chinese Academy of Fishery Sciences, Guangzhou, China
5 Shenzhen Key Lab of Marine Genomics, Guangdong Provincial Key Lab of Molecular Breeding in Marine Economic Animals, BGI Academy of Marine Sciences, BGI Marine, Shenzhen, Guangdong 518081, China

**Submitted:** 26 February 2025

\* Corresponding authors. E-mail: muxd@prfri.ac.cn; shiqiong@genomics.cn; shiqiong@szu.edu.cn

† Contributed equally.

Preprint submitted at https://doi.org/10.1101/2025.02.27.640546

## ABSTRACT

*Sinocyclocheilus*, a genus of tetraploid fishes endemic to Southwest China's karst regions, are classified as second-class nationally protected species due to their fragile habitat. Limited high-quality genomic resources have hampered studies on their phylogenetic relationships and the origin of their polyploidy. Here, we present a high-quality genome assembly of the most abundant *Sinocyclocheilus* species, the golden-line barbel (*Sinocyclocheilus grahami*), by integrating PacBio long-read and Hi-C sequencing. The resulting scaffold-level genome-assembly is 1.6 Gb long, with a scaffold N50 of up to 30.7 Mb. We annotated 42,806 protein-coding genes. Also, 93.1% of the assembled genome sequences (about 1.5 Gb) and 93.8% of the total predicted genes were successfully anchored onto 48 chromosomes. Furthermore, we obtained chromosome-level genome assemblies for four other *Sinocyclocheilus* species (*S. anophthalmus*, *S. maitianheensis*, *S. anshuiensis*, and *S. rhinocerous*) based on homologous comparisons. These genomic resources will enable in-depth investigations on cave adaptation, improvement of economic values, and conservation of diverse *Sinocyclocheilus* fishes.

**Subjects** Genetics and Genomics, Marine Biology, Animal Genetics

## INTRODUCTION

*Sinocyclocheilus* (order: Cypriniformes; family: Cyprinidae; subfamily: Barbinae) is a genus of tetraploid fishes endemic to the karst regions of the Yunnan-Guizhou plateau and surrounding areas in Southwest China, including Guangxi, Guizhou, Yunnan, and the Hubei provinces [1]. All members in this genus are classified as second-class protected species, highlighting the urgent need for their conservation and further investigation. Despite recent efforts in research and development, such as an artificial breeding program for

*S. yunnanensis* to prevent extinction [2], many other species, particularly those with small populations and limited distributions, remain in a threatened status.

Due to the long-term geographic isolation, *Sinocyclocheilus* species have undergone significant speciation, making it the most species-rich genus of cavefish, with 76 known members [1]. This genus inhabits various ecological environments, ranging from surface-dwelling to semi-cave-dwelling and cave-restricted. These distinct habitat types lead to diverse traits in morphology, behavior, and physiology [3], making them good models for studying cave adaptation and phylogenetic evolution. Although *Sinocyclocheilus* is of significant scientific interest, high-quality genomic resources and whole genome-based comparative studies are rare among *Sinocyclocheilus* fishes. The lack of genomic information hinders a deeper understanding of key evolutionary questions, such as phylogenetic relationships, the origin of polyploidy, and the evolution of ancestral chromosomes within this genus.

To enrich the genetic resources for *Sinocyclocheilus* members, we constructed a chromosome-level genome assembly for the most abundant and surface-dwelling representative, *S. grahami* (NCBI:txid75366, locally named golden-line barbel), using PacBio and Hi-C sequencing technologies, and subsequently conducted a homologous comparison to obtain chromosome-level genome assemblies for four other *Sinocyclocheilus* species: surface-dwelling *S. maitianheensis* (NCBI:txid307951), semi-cave-dwelling *S. rhinocerous* (NCBI:txid307959), cave-restricted *S. anshuiensis*, (NCBI:txid1608454), and *S. Anophthalmus* (NCBI:txid307955). To verify the allotetraploid origin of *Sinocyclocheilus*, we conducted a genome-wide synteny analysis between *S. grahami* and its relative, the common carp (*Cyprinus carpio*). The analysis revealed extensive chromosomal rearrangements and supported the allotetraploid origin of the *Sinocyclocheilus* genus. The genomic data we present in this paper provide valuable genetic resources for a deeper investigation into the mechanisms of cave adaptation, and for exploring the potential economic and ecological values of various *Sinocyclocheilus* species.

## METHODS

### Sample collection, DNA extraction, and genome and transcriptome sequencing

A muscle sample of artificially bred *S. grahami* was collected from the Endangered Fish Conservation Center of Kunming Institute of Zoology, which is located in Kunming City, Yunnan Province, China. Genomic DNA (gDNA) and total RNA were extracted using the Nucleic Acid Kit (Qiagen, Germantown, MD, USA) and TRIzol Reagent (Invitrogen, Carlsbad, CA, USA), respectively, following the manufacturer's instructions.

Multiple sequencing strategies were applied to construct a whole-genome assembly of *S. grahami*. In brief, the draft genome assembly based on Illumina sequencing technology (Illumina Inc., San Diego, CA, USA) was obtained using our previous study [4] as a reference. The genomic DNA from muscle tissue in our present study was used to construct a SMART bell library with an insert size of 20 kb, and this library was subsequently sequenced on a PacBio Sequel platform (Pacific Biosciences, Menlo Park, CA, USA). For the construction of a chromosome-level genome assembly, a Hi-C (High-throughput chromosome conformation capture) library was generated for sequencing on an Illumina HiSeq X-Ten platform. In addition, a paired-end library with an insert size of 400 bp was constructed from the



extracted gDNA and then sequenced on an Illumina HiSeq X-Ten platform for genome size estimation. Adapters, duplicated reads, and low-quality reads with 10 or more N bases were removed by SOAPfilter v2.2 [5]. For transcriptome sequencing (to support the annotation of genes), a paired-end library with an insert size of 350 bp was generated and then sequenced on an Illumina HiSeq X-Ten platform. Raw data were filtered by SOAPnuke v1.0 (RRID:SCR_015025) [6].

## Genome size estimation and chromosome-level genome assembly

A 17-mer frequency distribution, confirmed to be a Poisson pattern [7], was applied to estimate the genome size of *S. grahami* with a library of short-inserted size (400 bp). The genome size calculation formula was set as follows [4]: Genome Size = $K_{num}/K_{depth}$ ($K_{num}$ is the number of 17-mer; $K_{depth}$ is the sequencing depth at the core peak frequency).

Based on our published contigs of *S. grahami* [4], we performed a hybrid genome assembly by combining short contigs [4] with PacBio long reads into the primary scaffolds by using DBG2OLC v1.1 [8] with defaulted parameters. These scaffolds were subsequently extended using SSPACE v2.0 (RRID:SCR_005056) [9]. Minimap (RRID:SCR_018550) [10] and Racon (RRID:SCR_017642) [11] were employed for two rounds of error correction to obtain the final scaffolds with the assistance of PacBio long reads.

The Hi-C raw reads were mapped onto these scaffolds by Bowtie2 (RRID:SCR_016368) [12], and quality control was performed using HiC-Pro v2.8.0 (RRID:SCR_017643) [13] to obtain data for generating a genome-wide interaction matrix. Juicer v1.5 (RRID:SCR_017226) [14] and 3D-DNA *de novo* v170123 [15] were used to arrange and orient scaffolds into chromosomes. A Hi-C heatmap was drawn by HiCPlotter v0.6.6 [16] for visualization.

## Annotation of repeat sequences, gene and function

Three prediction methods were combined for the annotation of repeat sequences, including *de novo*, homolog-based, and tandem repeat prediction. A *de novo* repeat library was built using RepeatModeler v1.04 (RRID:SCR_015027) [17] and LTR_FINDER v1.0.6 (RRID:SCR_015247) [18]. Genome sequences were mapped onto this library to identify repeat sequences using RepeatMasker v4.06 (RRID:SCR_012954) [19]. For homolog-based predictions, transposable elements were identified using RepeatMasker v4.06 and RepeatProteinMask v4.06 [19] based on the Repbase TE v21.01 (RRID:SCR_021169) [20] library. Tandem repeat sequences were finally identified by Tandem Repeats Finder v4.09 (TRF, RRID:SCR_022065) [21].

Protein-coding genes were predicted by integrating three methods: homology-based annotation, *de novo* prediction, and transcriptome-based annotation. Protein sequences of five representative teleost species were downloaded from NCBI [22] for genome-wide mapping onto *S. grahami*: zebrafish (*Danio rerio*, NCBI accession: GCF_000002035.6), medaka (*Oryzias latipes*, NCBI accession: GCF_002234675.1), *S. anshuiensis*, *S. rhinocerous* and *S. grahami* (the primary genome assemblies using Illumina data). BLAT (RRID:SCR_011919) [23] and GeneWise v2.4.2 (RRID:SCR_015054) [24] were used for sequence alignment and gene structure prediction. Augustus v3.2.1 (RRID:SCR_008417) [25] was used to *de novo* predict coding sequences (CDS) after the repeat elements were masked. Hisat v0.1.6 (RRID:SCR_015530) [26] and Cufflinks v2.2.1 (RRID:SCR_014597) [27] were employed to perform the transcriptome-based annotation. Finally, a non-redundant gene set was merged by MAKER v2.31.8 (RRID:SCR_005309) [28]. For function annotation, we



searched four public databases (Swiss-Prot [29], TrEMBL [29], InterPro [30] and KEGG [31]) as references to complete the annotation of gene functions.

## Pseudochromosome construction of another four scaffold-level assemblies of different *Sinocyclocheilus* fishes

The general chromosome number of *Sinocyclocheilus* fishes is 96 [32]. Pairwise whole-genome alignments were used to construct pseudochromosomes of scaffold-level assemblies for four *Sinocyclocheilus* fishes (using data from three previously published and one unpublished genome assemblies): *S. anshuiensis* (GCF_001515605.1), *S. rhinocerous* (GCF_001515625.1) [4], *S. maitianheensis* (GCA_018148995.1) [33] and *S. anophthalmus* (GCA_044706345.1) [34] based on the reference chromosome-level assembly of *S. graham* [4].

 Lastz v1.1 (RRID:SCR_018556) [35] was used to process the genome alignments. Those aligned sequences with a length of more than 10 kb were retained for pseudochromosome construction. Synteny blocks of each genome for all five *Sinocyclocheilus* members were identified using MCScanX (RRID:SCR_022067) [36] after self-aligning with their own protein-set using BLAST [37] and the optimized parameter of *E*-value set to less than $1 \times 10^{-5}$. Circos figures were plotted using Circos (RRID:SCR_011798) [38].

## Subgenome identification in *S. grahami* and phylogenetic analysis

The common carp (*Cyprinus carpio*) is a well-known allotetraploid species [39] that shared a recent genome-wide duplication event with *Sinocyclocheilus* species, as we reported [33]. Subgenomes A and B of the common carp [40] were used as the references to identify corresponding synteny blocks in the genomes of goldfish (*Carassius auratus*), *S. graham*, and *S. anophthalmus* for subsequent subgenome construction using MUMmer v4.0beta1 (RRID:SCR_018171) [41]. RectChr (RRID:SCR_026859) [42] was used to visualize the synteny blocks and chromosome structure variations.

 For the phylogenetic analysis, BLASTp (RRID:SCR_001010) [37] and OrthoMCL (RRID:SCR_007839) [43] were used for protein sequence alignment and gene family clustering. All the single-copy orthologous genes were aligned using MUSCLE v3.8.31 (RRID:SCR_011812) [44] for all genomes and subgenomes. Subsequently, Gblocks (RRID:SCR_015945) [45] was used to obtain the conservative multi-sequence alignments. Finally, we used PhyML v3.0 (RRID:SCR_014629) [46] to construct a phylogenetic tree using the maximum likelihood method. MCMCtree in the PAML package (RRID:SCR_014932) [47] was used to estimate the divergence time from the above-mentioned fishes and other representative species.

## RESULTS AND DISCUSSION

## Chromosome-level genome assemblies of the five *Sinocyclocheilus* species

A total of 86.4 Gb, 79.1 Gb, and 229.0 Gb of Illumina, Pacbio, and Hi-C reads, respectively, were sequenced. We constructed a chromosome-level genome assembly for *S. grahami* using PacBio and Hi-C sequencing technologies. For the K-mer analysis, we estimated the genome size to be 1.9 Gb. The final chromosome-level genome assembly of *S. grahami* is 1.6 Gb with a contig N50 of 738.5 kb and a scaffold N50 of 30.7 Mb. About 93.1% of the assembled genome sequences (1.5 Gb) and 93.8% of the predicted genes were anchored

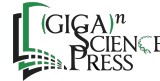

**Table 1.** Statistics of repeat sequences among the *S. grahami* genomes.

| Type | S. grahami | |
|---|---|---|
| | **Repeat Size (bp)** | **% of genome** |
| ProteinMask | 72,493,669 | 4.6 |
| RepeatMasker | 330,527,709 | 20.8 |
| TRF | 33,729,084 | 2.1 |
| *De novo* | 372,094,752 | 23.4 |
| Total | 583,165,599 | 36.7 |
| DNA | 374,708,178 | 23.6 |
| LINE | 105,180,165 | 6.6 |
| SINE | 5,429,507 | 0.3 |
| LTR | 145,884,791 | 9.2 |
| Other | 4,064 | 0 |
| Unknown | 2,578,927 | 0.2 |
| Total | 547,757,670 | 34.5 |

**Table 2.** Protein-coding gene annotation of *S. grahami* genome.

| Method | Software or Species | Gene number | Average | | | | |
|---|---|---|---|---|---|---|---|
| | | | Transcript Length (bp) | CDS Length (bp) | Exons per Gene | Exon Length (bp) | Intron Length (bp) |
| *de novo* | Augustus | 47,723 | 20,378 | 1,148 | 7.6 | 150.9 | 2,911 |
| Homolog | *Danio rerio* | 64,268 | 36,029 | 1,523 | 11.4 | 134.1 | 3,333 |
| | *Oryzias latipes* | 29,532 | 29,701 | 1,637 | 12.5 | 131.2 | 2,445 |
| | *Sinocyclocheilus anshuiensis* | 55,080 | 19,982 | 1,644 | 12.6 | 130.3 | 1,579 |
| | *Sinocyclocheilus rhinocerous* | 57,118 | 21,844 | 1,631 | 12.5 | 130.6 | 1,760 |
| | *Sinocyclocheilus grahami* | 49,556 | 29,535 | 1,507 | 11.1 | 135.5 | 2,768 |
| Transcriptome | | 31,114 | 9,515 | 1,628 | 7.6 | 214.7 | 1,198 |
| Consensus | MAKER | 42,806 | 18,370 | 1,331 | 8.9 | 148.2 | 1,984 |

Note: In homolog annotation, we used the genome and gene set of *S. grahami*, which were assembled using Illumina data [46].

**Table 3.** The number of functional assignments from diverse databases.

| | **Number** | **Percentage (%)** |
|---|---|---|
| **Total** | 42,806 | 100 |
| **InterPro** | 29,358 | 68.6 |
| **KEGG** | 34,734 | 81.1 |
| **Swissprot** | 33,908 | 79.2 |
| **TrEMBL** | 38,498 | 89.9 |
| **Annotated** | 39,458 | 92.2 |
| **Unanotated** | 3,348 | 7.8 |

onto 48 chromosomes (Figure 1A–B). For the BUSCO result (RRID:SCR_015008), 85.0% (3,093) of the BUSCO genes were complete, with 60.6% (2,205) being identified as single-copy, 24.4% (888) as duplicated, and a mere 3.6% (131) as fragmented.

A total of 583.2 Mb repeat sequences were annotated (Table 1). A sum of 42,806 protein-coding genes were annotated from the *S. grahami* genome assembly (Table 2), and 39,458 (92.2% of all) genes were annotated with functions. The detailed function results are shown in Table 3. We also constructed chromosome-level genome assemblies for the other four *Sinocyclocheilus* species based on homologous comparisons (Figure 1C). Over 82% of the genome sequences of all four species were anchored on these constructed chromosomes.

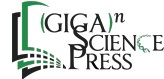

**Figure 1.** (A) Circos atlas of the reference chromosome-level genome assembly of *S. grahami*. Rings from outside to inside show chromosome length (Mb), distribution of gene density in each 100-kb non-overlapping genomic window, distribution of SNP density in each 100-kb non-overlapping genomic window, GC content in each 100-kb non-overlapping genomic window, and internal syntenic blocks of chromosomes that were connected by green lines. Red lines mark a special syntenic block between chromosome 1 and chromosome 3. (B) Genome-wide Hi-C heatmap of the *S. grahami* genome. (C) Circos atlases of the chromosome-level genome assemblies of four *Sinocyclocheilus* species. (D–F) Two chromosomal fusions, five chromosomal translocations, and eighteen chromosomal inversion events between *S. grahami* (top) and *C. carpio* (bottom). (G) A phylogenetic tree of seven vertebrate genomes and eight sub-genomes of tetraploid species. The orange box represents the clade of sub-genome A; the blue box marks the clade of sub-genome B; the purple box highlights a clade homologous to the ancestors of sub-genome A. Diverge time is numbered in blue, and a geographic time scale in million-years-ago is provided.

## Allotetraploid origin of diverse *Sinocyclocheilus* members

To confirm that *Sinocyclocheilus* fishes originated from allotetraploid, we performed a genome-wide synteny analysis of *S. grahami* and *C. carpio* (Figures 1D–F and 2). Compared

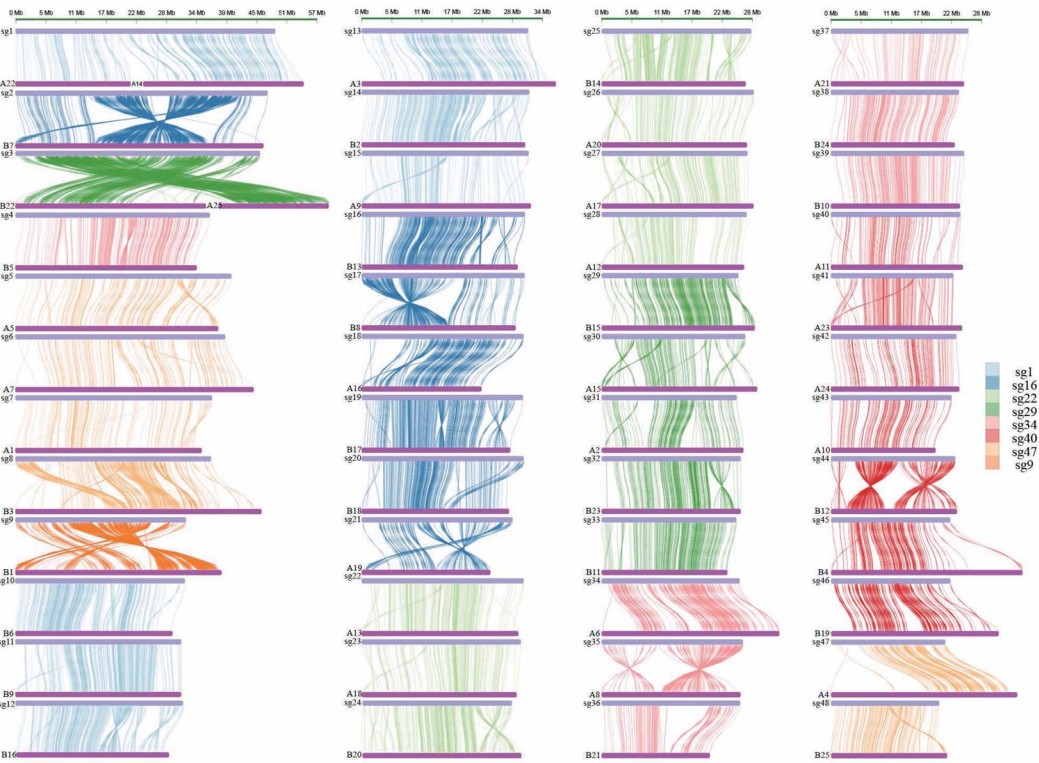

**Figure 2.** Genome synteny of *S. grahami* (top) and *C. carpio* (bottom).

with common carp, 18 large chromosomal rearrangements were observed in the *S. grahami* genome, including two chromosomal fusions (Figures 1D and 2), five chromosomal translocations (Figures 1E and 2), and eighteen chromosomal inversions (Figures 1F and 2). Among them, chromosome 1 of *S. grahami* was homologous to chromosomes A22 and A14 of the common carp; chromosome 3 of *S. grahami* was homologous to chromosomes B22 and A25 of the common carp. These alignments resulted in *S. grahami* having two fewer chromosomes than the common carp.

According to our synteny results, we renumbered the chromosomes of *S. grahami* and divided them into two sub-genomes. The other four *Sinocyclocheilus* members and goldfish were also identified with two sub-genomes for phylogenetic analysis. In the established phylogenetic tree, the group of sub-genome A was clustered into a single branch; the branch of subgenome B was homologous to the ancestors of *O. macrolepis*, *P. huangchuchieni*, and *P. tetrazona* (Figure 1G), similarly to patterns from early reports [38, 48].

## CONCLUSION

We constructed chromosome-level genome assemblies of five *Sinocyclocheilus* species. These reference genomics data are valuable resources for in-depth studies on phylogenetic evolution and biodiversity of various *Sinocyclocheilus* species, and lay a solid foundation for understanding cave adaptation and cavefish biology. Our current study can also contribute to species conservation and the exploitation of potential economic and ecological values of diverse *Sinocyclocheilus* members.

## DATA AVAILABILITY

The genome assembly of *S. grahami* was uploaded to NCBI under the BioProject PRJNA1172646, and the genome assembly of *S. anophthalmus* is available under the BioProject PRJNA669129. The Pacbio, HiC, and transcriptome reads are deposited in NCBI with accession numbers SRR32815372, SRR32815371, and SRR32815370, respectively. All other data, including the repeat and gene annotations, have been shared via the GigaDB repository [49], with separate entries for the individual species genomes [34, 50–53].

## ABBREVIATIONS

CDS, coding sequences; gDNA, Genomic DNA; LINE, long interspersed nuclear element; LTR, long terminal repeat; SINE, short interspersed nuclear element; TRF, Tandem Repeat Finder.

## DECLARATIONS

### Ethics approval and consent to participate

The authors declare that ethical approval was not required for this type of research.

### Competing interests

The authors declare that they have no competing interests.

### Authors' contributions

QS and JY conceived the study and designed the project. YO and JY managed the project and prepared samples. CB, RL, YO, and XM performed data analysis and wrote the manuscript. QS and JY revised the manuscript. All authors contributed to data interpretation.

### Funding

This study was supported by the National Key Research and Development Program of China (2023YFE0205100), Shenzhen Natural Science Foundation (no. JCYJ20241202124511016), Key Laboratory of Tropical and Subtropical Fishery Resources Application and Cultivation, Ministry of Agriculture and Rural Affairs, Pearl River Fisheries Research Institute, Chinese Academy of Fishery Sciences, Guangzhou, 51038, PR China (20220202).

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
