## [Editor Report]

Editor’s AssessmentSinocyclocheilus are a genus of freshwater cavefish fish that are endemic to the Karst regions of Southwest China. Having diverse traits in morphology, behavior, and physiology typical of cavefish, that make them interesting models for studying cave adaptation and phylogenetic evolution. The manuscript assembled chromosomal-level genomes of five Sinocyclocheilus species, and conducted allotetraploid origin analysis on these species. Assembling S. grahami (the golden-line barbel), using PacBio and Hi-C sequencing technologies, a final chromosome-level genome assembly was 1.6 Gb in size with a contig N50 of 738.5 kb and a scaffold N50 of 30.7 Mb. With 93.1% of the assembled genome sequences and 93.8% of the predicted genes anchored onto 48 chromosomes. Subsequently the authors conducted a homologous comparison to obtain chromosome-level genome assemblies for four other Sinocyclocheilus species: S. maitianheensis, S. rhinocerous, S. anshuiensis, and S. Anophthalmus. With over 82% of the genome sequences anchored on these constructed chromosomes. Peer review provided clarification on the assembly strategy and provided more benchmarking. This data having the potential to contribute to species conservation and the exploitation of potential economic and ecological values of diverse Sinocyclocheilus members.Editor’s AssessmentSinocyclocheilus are a genus of freshwater cavefish fish that are endemic to the Karst regions of Southwest China. Having diverse traits in morphology, behavior, and physiology typical of cavefish, that make them interesting models for studying cave adaptation and phylogenetic evolution. The manuscript assembled chromosomal-level genomes of five Sinocyclocheilus species, and conducted allotetraploid origin analysis on these species. Assembling S. grahami (the golden-line barbel), using PacBio and Hi-C sequencing technologies, a final chromosome-level genome assembly was 1.6 Gb in size with a contig N50 of 738.5 kb and a scaffold N50 of 30.7 Mb. With 93.1% of the assembled genome sequences and 93.8% of the predicted genes anchored onto 48 chromosomes. Subsequently the authors conducted a homologous comparison to obtain chromosome-level genome assemblies for four other Sinocyclocheilus species: S. maitianheensis, S. rhinocerous, S. anshuiensis, and S. Anophthalmus. With over 82% of the genome sequences anchored on these constructed chromosomes. Peer review provided clarification on the assembly strategy and provided more benchmarking. This data having the potential to contribute to species conservation and the exploitation of potential economic and ecological values of diverse Sinocyclocheilus members.

---

## [Reviewer Report]

Reviewer name and names of any other individual's who aided in reviewer Jun WangDo you understand and agree to our policy of having open and named reviews, and having your review included with the published papers. (If no, please inform the editor that you cannot review this manuscript.)YesIs the language of sufficient quality?YesPlease add additional comments on language quality to clarify if needed
Are all data available and do they match the descriptions in the paper? YesAdditional CommentsAre the data and metadata consistent with relevant minimum information or reporting standards? See GigaDB checklists for examples <a href="http://gigadb.org/site/guide" target="_blank">http://gigadb.org/site/guide</a>YesAdditional CommentsIs the data acquisition clear, complete and methodologically sound?YesAdditional CommentsIs there sufficient detail in the methods and data-processing steps to allow reproduction?YesAdditional CommentsIs there sufficient data validation and statistical analyses of data quality? YesAdditional CommentsIs the validation suitable for this type of data?YesAdditional CommentsIs there sufficient information for others to reuse this dataset or integrate it with other data?YesAdditional CommentsAny Additional Overall Comments to the AuthorThe manuscript assembled chromosomal-level genomes of five Sinocyclocheilus species, and conducted allotetraploid origin analysis on these species. The manuscript was meaningful and provided valuable genome resources in Sinocyclocheilus genus, which will further help with the evolution and functional genomics of these species. The analysis was accurate, and the results were solid. My comments are as follows 1. Please detail the method how you assembled four other species on homologous comparison? You just map the assembled scaffold to the reference genome? 2. In the manuscript, the author only provide the sequencing info of S.grahami but not other four species. What are the sequencing information of other four species, like how many reads have been sequenced for Illumina? 3. There was no results description for figure 2 and why there are only repeat annotation results for S.grahami no other four species?RecommendationMinor Revision

---

## [Reviewer Report]

Reviewer name and names of any other individual's who aided in reviewer Fei Li, Shili LiDo you understand and agree to our policy of having open and named reviews, and having your review included with the published papers. (If no, please inform the editor that you cannot review this manuscript.)YesIs the language of sufficient quality?YesPlease add additional comments on language quality to clarify if needed
Are all data available and do they match the descriptions in the paper? YesAdditional CommentsAre the data and metadata consistent with relevant minimum information or reporting standards? See GigaDB checklists for examples <a href="http://gigadb.org/site/guide" target="_blank">http://gigadb.org/site/guide</a>YesAdditional CommentsIs the data acquisition clear, complete and methodologically sound?YesAdditional CommentsIs there sufficient detail in the methods and data-processing steps to allow reproduction?YesAdditional CommentsIs there sufficient data validation and statistical analyses of data quality? YesAdditional CommentsIs the validation suitable for this type of data?YesAdditional CommentsIs there sufficient information for others to reuse this dataset or integrate it with other data?YesAdditional CommentsAny Additional Overall Comments to the AuthorThis paper entitled “Chromosome-level genome assemblies of five Sinocyclocheilus species” reported a chromosome-level golden-line barbel genome by using combination of Pacbio and Hi-C data. Using this chromosome-level assembly as reference, the author also constructed other four psedo chromosome-level assemblies of S.anophthalmus, S. maitianheensis, S. anshuiensis, and S. rhinocerous. These data are really important resource for conservation of these endanger species. However, some important results have not shown: 1. Protein BUSCO result has not been shown. 2. Raw reads were not uploaded to NCBI. 3. What’s the detailed number for functional annotation. Some minor suggestions: Add “,” before “and conservation”. What’s meaning of “R & D”? Line 58, “a good model” should be “good models”. Line 64, remove “at first”. Line 84, change “a” to “the”. Line 90, change ‘muscle’ to “muscle tissue”. Line 105, remove ‘which was’. Line 112, remove ‘this study’. Line 122, change “Repeat annotation, gene prediction, and function prediction” to “Annotation of repeat, gene and function”. Line 137, ‘with’ should be ‘by using’. Line 127, remove ‘(TEs)’. Line 134, What’s meaning of NCBI GenBank? Remove GenBank. Line 140, ‘was’ should be ‘were’. Line 178, ‘Species’ should be ‘species’.RecommendationMinor Revision